# Effects of Population Knowledge, Perceptions, Attitudes, and Practices on COVID-19 Infection Prevention and Control in NUST

**DOI:** 10.3390/ijerph19105918

**Published:** 2022-05-13

**Authors:** Caitlin Bosch, Edwig Hauwanga, Beauty E. Omoruyi, Benjamin I. Okeleye, Vincent I. Okudoh, Yapo G. Aboua

**Affiliations:** 1Department of Health Sciences, Faculty of Health and Applied Sciences, Namibia University of Science and Technology, Windhoek 9000, Namibia; caitlin.bosch@gmail.com (C.B.); ehauwanga@nust.na (E.H.); yaboua@nust.na (Y.G.A.); 2Applied Microbial and Health Biotechnology Institute, Cape Peninsula University of Technology, Cape Town 8000, South Africa; omoruyie@cput.ac.za; 3Department of Biotechnology and Consumer Science, Cape Peninsula University of Technology, Cape Town 8000, South Africa; ben_okeleye2005@yahoo.com

**Keywords:** COVID-19 pandemic, knowledge, perception, attitude, infection prevention, control

## Abstract

The global COVID-19 pandemic has had a major impact on the education sector of most countries. One of the basic CDC prevention guidelines is the implementation of non-pharmaceutical interventions (NPIs) to protect the health of students and staff members to curve the spread of COVID-19. The current study aimed to examine the knowledge, perceptions, attitudes, and practices of students at the Namibia University of Technology toward the COVID-19 pandemic. A cross-sectional descriptive survey was conducted using a closed-ended questionnaire. Data were collected from full-time students who were on campus during the COVID-19 pandemic between 29 January to 14 February 2021. The average knowledge about the modes of transmission, protective measures, and clinical symptoms ranged from 78% to 96%. About 31% of student respondents believed the virus was created in a laboratory, and 47% believed the vaccine has negative side effects and therefore, refused to take it. The three main sources of information about COVID-19 were social media (75%), television (63%), and friends and family (50%). The students had an overall positive attitude towards the implementation of NPIs. However, the importance of vaccine safety must be emphasized. Lockdowns should be lifted gradually to reduce the amount of time students are spending on online content. Reopening of classrooms for face-to-face study will bring unquestionable benefits to students and the wider economy.

## 1. Introduction

Pandemics are large-scale outbreaks of deadly viruses that have impacted all sectors of our society, including institutions [1]. Several viruses, including the influenza virus (H1N1 swine flu), filovirus (Ebola), flavivirus (Zika), and coronaviruses (SARS-CoV, MERS-CoV) have been discovered [1]. The recent virus (COVID-19) was named SARS-CoV-2 because its RNA genome shares approximately 80% similarity with other coronaviruses that cause the common cold and flu [2]. COVID-19 was formally declared a ‘pandemic’ in 2020 by the World Health Organization [3] due to the rapid spread of the disease worldwide. In February 2021, the confirmed cases had reached 111,593,583, with mortality death rates of 2,475,020 worldwide. In South Africa, confirmed cases were over 3,665,032, with death rates of 98,868. COVID-19 in Botswana had reached 262,652, with a death toll of 2614. In Namibia, a total of 37,896 cases, 411 deaths, and a recovery rate of 93.46% were confirmed. The proportion of confirmed COVID-19 cases in Namibia over the total confirmed cases in Africa was 1.36% [4]. Transmission of the virus spreads among humans via mucus or saliva droplets from direct infected contact with another through talking, coughing, or sneezing [5,6]. Symptoms of COVID-19 infection range from mild symptoms of fever, dry cough, and dyspnoea to significant hypoxia (difficulty in breathing) with acute respiratory distress syndrome (ARDS) [7]. People with underlying conditions, such as hypertension, HIV/AIDS, diabetes mellitus, cardiovascular disease, chronic pulmonary disease, chronic kidney disease malignancy, and chronic liver diseases are most susceptible [7].

When Namibia reported its first confirmed COVID-19 cases on 13 March 2020, the government implemented a national health emergency coordination committee under the Ministry of Health and Social Services (MOHSS) in collaboration with the National Public Health Emergency Operation Center (NPHEOC) and World Health Organization (WHO) to foresee all COVID-19 responders at the national level [4]. To limit the spread of infection during the first wave, the government declared a state of emergency on 17 March 2020 and immediately opted for several public health and safety lockdown measures. Early measures included a ban on international travel and mass gatherings and mandatory quarantine. From mid-April, the non-pharmaceutical interventions (NPIs) were implemented following WHO’s guidelines. NPIs were the most accessible interventions during the disease outbreak as vaccines or antiviral drugs were not available [8]. The purpose of implementing NPIs during the pandemic was to maintain a steady state of low-level transmission and mortality rate. The individual NPI protective measures implemented included social distancing, respiratory hygiene, and hand hygiene. The social distance of 1.5 m can reduce the transmission of the virus by limiting the number and amount of time individuals socialize [9]. The use of a face mask was mandatory, especially when crowded/enclosed settings cannot be avoided [9]. Hand hygiene was recommended in all settings, including healthcare and community settings for the prevention of COVID-19 [10]. Regular handwashing with soap and water for at least 20 s or hand sanitizing with alcohol-based hand sanitizer were also recommended.

The second level of protective NPIs that had been remarkably effective were the environmental NPIs, which included surface cleaning, disinfection, and optimal ventilation [10]. Regular disinfection using standard detergents on frequently touched surfaces, such as door handles, hand railings, buttons, or public transport was mandatory to reduce transmission of the virus. Proper ventilation to allow air exchange has been observed to decrease the transmission of the virus in indoor spaces and therefore, should be maintained. The third protective level implemented was the population/community NPIs, which included self-isolation of sick individuals and school closures. Isolation of symptomatic cases in hospitals or at home is an effective measure for reducing COVID-19 transmission [10]. In any educational setting where learners or staff are infected with the virus and to avoid further transmission, school/universities are allowed to shut down for proper disinfection. However, the support of NPIs varied across countries. What works for one country may be insignificant or ineffective in another country. Therefore, decisions about the implementation of NPIs require flexibility [11].

Previous knowledge, attitudes, and practices of university students towards seasonal and pandemic influenza have been examined [12]. Most of the participants’ responses to NPI measures have been extremely difficult to comply with. For example, in an online survey conducted on students and staff at a public university during the 2009 H1N1 swine flu pandemic, only about 64.9% of respondents accepted and complied with the adoption of NPI measures for self-protection [13]. Adherence to various COVID-19 preventive and health-related behaviors among frontline workers has been reported to be 56% in Prato Province, Italy [11]. The overall level of e-health scores comprising functional, communicational, critical literacy, and preventive behaviors related to COVID-19 among undergraduate students in healthcare in South Korea were observed to be 3.62 and was found to correlate with students’ responses to preventive measures [14]. The present study has implemented these NPI protective measures to prevent the spread of COVID-19 around the campus; hence, this study examines the students’ knowledge, perception, attitude, and practices toward the COVID-19 pandemic. The results obtained will be used to determine if there are improvements in adopting NPIs measures and if they should be implemented for any future disease outbreaks.

## 2. Materials and Methods

### 2.1. Study Design

A cross-sectional descriptive survey design was used in this study [15]. Closed-ended questionnaires were distributed online to all the full-time students who have access to the internet and WhatsApp. Students were examined based on their knowledge, perceptions, attitudes, and measures implemented toward the coronavirus disease (COVID-19) pandemic. The Namibia University of Science and Technology (NUST) was chosen for this study because it is one of the most populated higher education institutions in the country. Data was collected between 29 January to 14 February 2021. The survey was divided into five sections, including (i): sociodemographic characteristics, such as age, sex, and faculty in which they are enrolled; (ii): main sources from which students received information about COVID-19, which include television, newspapers, social media sites, WhatsApp, educational posters, scientific websites and articles, healthcare workers, friends and family, university campus website; (iii): knowledge about the modes of transmission, protective measures, and the clinical symptoms of COVID-19; (iv): students’ perception about the “lockdown” effectiveness in controlling the spread of the disease and vaccination; (v): students’ attitude and practices towards the protective measures implemented by the university.

### 2.2. Ethical Consideration

Ethics approval was obtained in accordance with the declaration of Helsinki by the Namibia University of Science and Technology ethics committee (FHAS-REC-14/2020). The research was purely based on observations without any participant names mentioned. Participants were informed that the survey was anonymous and voluntary, and data were treated confidentially.

### 2.3. Inclusion Criteria and Exclusion Criteria

Full-time students enrolled at the university that has been on campus during the COVID-19 pandemic were included. Part-time and distance students enrolled at the university during the COVID-19 pandemic were excluded to avert biased results.

### 2.4. Study Population and Sample Size

The study population included full-time students enrolled at a university campus across Windhoek during the 2020–2021 academic year. The sample size was interrupted due to students not being able to afford WhatsApp data to participate in the survey. In addition, students were faced with an uncertain environment and health issues of infection during the pandemic and hence, refrained from participating. For this reason, we surveyed about 112 students who were willing to join the survey. The sample size was estimated by calculating the total number of full-time students enrolled in 2018 with a confidence level of 95% and a 5% margin of error. According to the university’s Annual Report of 2018, 11,235 students were enrolled of which 6179 (55%) were enrolled as full-time students. The sample size was calculated using Equation (1):(1)$$n=\frac{z^{2}pq}{d^{2}}$$
where: *n* = desired minimum sample size; *z* = 1.96 (at 95% confidence interval); *p* = 0.55 (55% = estimated number of full-time students enrolled at the university in 2018); *q* = 1 − *p*; *d* = 0.05 (5% margin of error). Hence, the estimated sample size was calculated as follows:(2)$$n=\frac{1.96^{2}\times 0.55\times \left(1-0.55\right)}{0.05^{2}}\;n=380$$

### 2.5. Data Analysis

The survey responses were imported into Google spreadsheets and analyzed using IBM^®^ SPSS version 26. Frequencies and percentages were used for the dichotomous and multiple response variables. The mean, standard deviation, and 95% confidence intervals were calculated for the nominal and ordinal data. Wilson’s score method, adapted from Weaver 2014, was used to calculate the confidence intervals. The student’s level of knowledge about the modes of transmission, the protective measures, and the clinical symptoms were measured using a scoring system of ‘1’ for a correct answer and ‘0’ for an incorrect or unsure answer. The maximum score of each section was 6, 8, and 9, respectively, with a total score of 23. The students’ perceptions of COVID-19 and their attitudes and practices toward the protective measures (non-pharmaceutical interventions) implemented were calculated using a 5-point Likert scale scoring system, where strongly agree = 5, agree = 4, neutral = 3, disagree = 2, strongly disagree = 1. The scoring system was reversed for the negatively worded questions, where strongly agree = 1, agree = 2, neutral = 3, disagree = 4, and strongly disagree = 5.

## 3. Results

### 3.1. Sociodemographic Characteristics

A total of 112 students responded to the online survey. Table 1 illustrates the sociodemographic characteristics of the students. Eighty (71.43%) students were female and thirty-two (28.57%) were male. Among the students, 103 (91.96%) were between the ages of 18–25 years, 9 (8.04%) were between the ages of 26–35 years, and none were above the age of 35 years.

Figure 1 illustrates the proportion of students according to the faculty in which they are enrolled. Students from the Faculty of Health and Applied Sciences were the highest with 66.07%, followed by the Faculty of Engineering with 13.39%, and the Faculty of Natural Resources and Spatial Sciences with 8.93%.

Figure 2 shows the sources from which the students received information about COVID-19. The four main sources of information identified were social media (75.00%), television (63.39%), friends and family (50.00%), and WhatsApp (49.11%). Less than half of the students reported receiving COVID-related information from newspapers (34.82%), educational posters (29.46%), scientific websites/articles (33.93%), healthcare workers (17.86%), university campus/websites (30.36%), and YouTube (0.89%).

The students’ levels of knowledge about COVID-19 were assessed in three different sections, using modes of transmission, protective measures, and clinical symptoms. For each section, the correct answers were expressed using frequencies and percentages. The mean score of each section was calculated by adding the mean values of each question. The average level of knowledge of each section was calculated by dividing the mean score by the total score and multiplying by 100%. Table 2 shows the answers for the modes of transmission. The most frequently reported modes of transmission were person to person via respiratory droplets (98.21%), followed by contact with contaminated surfaces (94.64%). Interestingly, only half of the students reported that COVID-19 cannot be transmitted through sexual intercourse or water while swimming. The average mean score about the modes of transmission of COVID-19 was 4.73 ± 1.031 (range, 0–6) with an average knowledge level of 78.83% ([4.73/6] × 100).

Table 3 unveils the answers for the protective measures against COVID-19. Overall, students had thorough knowledge about the protective measures against COVID-19. ‘Avoid touching your eyes, mouth, and nose’ received the highest score of 99.11%, whereas ‘Once you have had the disease you cannot get reinfected’ received the lowest score of 91.07%. The average mean score for the protective measures was 7.69 ± 0.601 (range, 0–8) with an average knowledge level of 96.13% ([7.69/8] × 100).

Table 4 unveils the answers to the clinical symptoms of COVID-19. Interestingly, less than half of the students knew that diarrhea (43.75%) and vomiting (43.75%) were symptoms of COVID-19. The average mean score for the clinical symptoms of COVID-19 was 7.21 ± 1.434 (range, 0–9) with an average knowledge level of 80.11% ([7.21/9] × 100). Overall, the students had a high level of knowledge about COVID-19 at 85.35% ([19.63/23] × 100).

### 3.2. Students’ Perceptions of COVID-19

The students were provided with general statements about COVID-19 to measure their level of perception. Their answers were rated on a scale from 1 = strongly disagree to 5 = strongly agree. The negatively worded statements were reversed where 5 = strongly disagree and 1 = strongly agree. The mean score of each statement was calculated. Interval data ranges were calculated by subtracting the maximum score from the minimum score (5 − 1 = 4) and dividing by the total number of points to get the interval range (4/5 = 0.8). The interval data ranges were used to determine the average perception level for each statement, whereby strongly disagree falls between the interval range 1.00–1.80, disagree falls between 1.81–2.60, neutral falls between 2.61–3.40, agree falls between 3.41–4.20, and strongly agree falls between 4.21–5.00. The interval ranges for the negatively worded questions were reversed, whereby strongly agree falls between the interval range of 1.00–1.80, agree falls between 1.81–2.60, neutral falls between 2.61–3.40, and disagree falls between 3.41–4.20, and strongly disagree falls between 4.21–5.00.

Table 5 shows the mean SD calculated for each statement regarding the students’ perception of COVID-19. The first statement has a mean of 3.92 ± 1.083 and falls between the interval range of 3.41–4.20. This demonstrates that most students agree that the lockdown was an effective method of controlling the spread of the disease. Whilst the second, third and fourth statements had mean values of 3.20 ± 1.400, 2.85 ± 1.117, and 3.11 ± 1.085, respectively, and hence fall between the interval range mean of 2.61–3.40. This demonstrated that the students were neutral to the statements.

Interestingly, most of the students disagree with the fifth statement ‘*I am willing to take the vaccination should any become available,*’ with a mean of 2.60 ± 1.372 (interval range, 1.81–2.60). The last question gave a mean of 2.63 ± 1.302 (interval range, 2.61–3.40), which suggests that most of the students were neutral to the question ‘*I believe the vaccine could have negative side effects and therefore, refuse to take it.*’ Overall, the students’ levels of perceptions of COVID-19 were neutral.

### 3.3. Attitudes and Practices of Students

The students’ attitudes and practices of protective measures implemented by the university were measured using the 5-point Likert scale, which was previously used for the students’ levels of perceptions. The mean score for each question was calculated and rated according to the interval data range to measure the students’ average levels of agreement toward the university’s implementation of NPIs. Table 6 shows the mean and SD calculated for each question. The first question has a mean of 4.54 ± 0.967 and falls between the interval range of 4.21–5.00, indicating that most students strongly disagree that the need for sanitization when entering a campus or a building is unnecessary. The second question has a mean of 4.76 ± 0.713 (interval range, 4.21–5.00), which also shows that the majority of the students strongly disagree with the statement ‘*Only sick people should wear masks*.’ The third question has a mean of 3.66 ± 1.205 (interval range, 3.41–4.20), showing that most of the students agree that the university provided them with sufficient COVID-related information. Overall, the students demonstrated a positive attitude towards the protective measures implemented by the university.

Table 7 shows the mean and SD calculated for each statement regarding the students’ practices of protective measures implemented by the university. The first statement has a mean of 4.54 ± 0.657 and falls between the interval range of 4.21–5.00, indicating that most students strongly agree to wear masks when they are on campus. The second statement has a mean of 3.42 ± 1.242 (interval range, 3.41–4.20), showing that most of the students disagree with ‘*I do not wear my mask when I am outside the classroom/building.*’, which contradicts the previous statement about wearing a mask on campus.

Surprisingly, the responses from ‘I practice social distancing on campus and in the classroom/auditorium.’ and ‘I do not practice social distancing on campus when I am with my friends.’ are contradictory as students disagree with the former statement (2.02 ± 0.838; interval range, 1.81–2.60), and then become neutral in the latter statement (2.73 ± 1.115; interval range, 2.61–3.40). The fifth question has a mean of 4.37 ± 0.684 (interval range, 4.21–5.00), which indicates most students strongly agree that they wash their hands more frequently. The last question has a mean of 4.23 ± 1.013 (interval range, 4.21–5.00), which indicates most students strongly disagree with ‘I attended class even when I had COVID-related symptoms.’

## 4. Discussion

The current cross-sectional study examined the knowledge, perception, attitude, and practices of full-time NUST students towards the COVID-19 pandemic. The limitations of the study were the fear of students facing an uncertain environment, the health shock of the pandemic may have interrupted the one-to-one survey, and a sample size of only 112 participants in the study instead of 380. This may be due to the public concern about the misinformation on the origin of the COVID-19 virus, which also affected the compliance of vaccine uptake negatively. In addition, most of the students were not able to afford the WhatsApp data to participate in the survey. However, the students that participated in the study showed an overall high level of knowledge about COVID-19 transmission, symptoms, and protective measures implemented by the university. Based on student responses, female participants and students from the faculty of health and applied sciences responded more; however, the students had an overall knowledge level score of 85.35%, which is expected as it is a year since the WHO declared COVID-19 as a pandemic. They also demonstrated a positive attitude towards the protective measures implemented by the university. However, they did not adhere to all the protective measures implemented. The findings of this study are comparable to other studies conducted amongst university students about COVID-19. Olaimat et al. [16] surveyed Jordan university students to determine their knowledge and information sources about COVID-19. When asked questions about the symptoms of COVID-19, around 55–71% of students were aware of the less common symptoms such as weakness (44.3%), runny nose (40.4%), vomiting (28.9%), and diarrhea (40.8%). This is in contrast with the results of this study where less than half of the students were aware that diarrhea (43.57%) and vomiting (31.25%) were symptoms of COVID-19. The reason could be that gastrointestinal symptoms are typically stated in scientific articles instead of on social media platforms. Nonetheless, the average knowledge level about the clinical symptoms of COVID-19 was 80.11%.

The students had an average knowledge level of 78.83% about COVID-19 modes of transmission. More than 90% were aware that COVID-19 can be transmitted through contact with contaminated surfaces, shaking hands, and respiratory droplets. University officials are making sure to maintain surface disinfection and good ventilation daily to curve the spread. The students had average knowledge of 96.13% about the protective measures against COVID-19. Expectedly, avoidance of touching your face and washing hands regularly had the highest scores of 99.11% and 98.21%. When asked to select the sources from which students had received information about COVID-19, 75.00% chose social media followed by television (63.39%). Only 33.93% chose scientific websites as a source of information. There appears to be a trend amongst university students worldwide. Singh et al. [17] conducted a study amongst university students in India. Social media (81.4%) and television (75.3%) were the main sources of receiving COVID-related information. About 34.3% sought information from healthcare workers, which is higher compared to the 17.86% in this study that received information from healthcare workers.

The students’ average levels of perceptions of COVID-19 were unremarkable. The lockdown as a method of controlling the spread of the disease was only agreed upon by 34.82% of student respondents. About 41% of students agreed that it is impractical to practice social distancing during everyday activities. Further results showed that only 25.89% of students are willing to take the vaccine. However, 47.32% believed the vaccine has negative side effects and therefore, refused to take it. A study in Jordan and Kuwait assessed the attitudes of the participants towards prospective COVID-19 vaccines. They observed that the high rate of vaccine hesitancy is associated with reliance on social media as the main source of information about the COVID-19 vaccines [18]. Even though the rate of social media access is high in this study, one cannot assume that the student’s refusal to take the vaccine is related to using social media as a source of information, as the relationship between the two variables has not been measured. Allington et al. [19] tested the relationship between conspiracy beliefs and the use of social media platforms. They observed that the relationship between social media platforms and conspiracy beliefs was statistically significant (*p* < 0.001). The strongest association with conspiracy beliefs was social media and Facebook. Thirty-one percent of student respondents in this study believe that the virus was created in a laboratory.

The students had an overall positive attitude towards the protective measures implemented by the university. However, only 5.36% agreed that hand sanitizing when entering campus/buildings is unnecessary. About 2.68% of students agree that only sick people should wear a mask to prevent the spread of COVID-19. This looks promising compared to the study conducted by Khasawneh et al. [20] in which 60.6% agreed that only sick people should wear a mask. The reason for the data differences could be that the Khasawneh et al. [20] study was conducted at the very beginning of the pandemic (March 2020), whereas this study was conducted in February 2021. It could also be assumed that the attitudes of students towards the implementation of NPIs has improved as more information about the pandemic became available.

Based on the present findings, the students did not practice all the protective measures implemented by the university. Over 90% agreed that they wear a mask when they are on campus, and 29.79% take off the mask when they are outside classes or during lunch break. This defeats the purpose, as students are exposed to being infected with the virus. By way of comparison, only 9.7% of medical students at Jordan university considered it necessary to wear masks often [20]. Nevertheless, a lack of adequate knowledge about the severity of the pandemic disease may hinder the compliance of mask wearing [21]. Temperature records were not included in the study because data records of students’ temperature taken at the entry points of the campus are never recorded. Additionally, the thermometers used are not calibrated; they are used until their life span is reached.

Contradictory results about social distancing were obtained. About 66% of students do not practice social distancing on campus, and 48.21% do not practice it with friends. It could be that it is the least strictly implemented measure, as students do not maintain social distancing on campus because the 1.5-m marks are hardly found at sitting areas. Secondly, the 1.5-m markings are only found at the gates of the institution, and students are not reminded by the university officials to maintain distance. Social distancing is very crucial, especially when people fail to wear their masks, as it limits the transmission of the respiratory droplets.

Most of the students were neutral to the questions, but the calculated frequency percentage yielded only 25.89%. Another main practice in preventing the spread of the virus is self-isolation. In this study, 81.25% of students agree that they never attended class when they had COVID-related symptoms. The sample size used in this study was small, though a convenient sampling method was used, whereby participants were selected based on accessibility and availability.

## 5. Conclusions

The study revealed how important it is to have accurate knowledge in relation to the mode of transmission, symptoms, and protective measures of infectious disease as the average knowledge rate was recorded in the range of 78–96%. This motivated the participants to comply with the use of face masks, as a high compliance rate was observed in this study. The findings of this study will be of great benefit to the government, department of health, and clinical facilities to prepare for a future pandemic. The most frequent source of information about COVID-19 was social media. Vaccine doses approved by the Food and Drug Administration (FDA), such as Pfizer, Johnson & Johnson, etc., should be prioritized more on social media for students and others to be aware of their safety, as they help the body develop an immune response against foreign-made viruses. Secondly, the school authorities should ensure that the vaccines, including other NPIs, such as alcohol-based hand sanitizer and soap, are readily available for everyone. The student’s insights about the lockdown are that it should be lifted gradually because it has cost students enormous amounts of time and money spent consuming online content. Students should be allowed to attend classes remotely, have inspiring conversations with faculty, collaborate with researchers in the laboratory, and experience social life on campus. Reopening classrooms for the face-to-face study will bring unquestionable benefits to students and the wider economy. It will also benefit families by enabling some parents to return to work. The future study should, therefore, focus on regular surveillance to monitor the maintenance or improvement of COVID-19 control and prevention and the general effect of the pandemic on teaching and learning.

## Figures and Tables

**Figure 1 ijerph-19-05918-f001:**
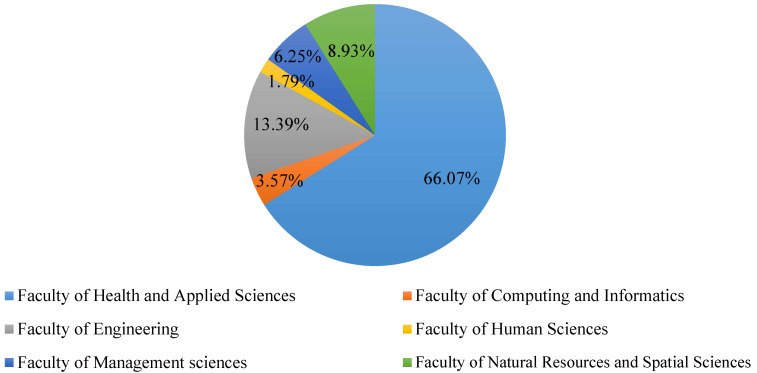
Faculties in which students have enrolled.

**Figure 2 ijerph-19-05918-f002:**
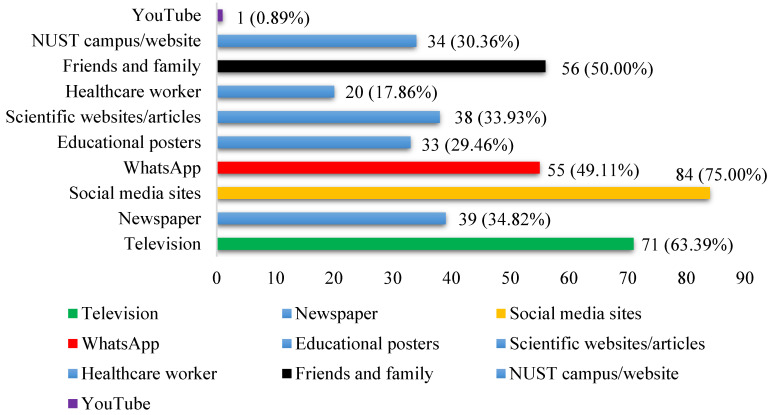
Sources of information about COVID-19 students’ levels of knowledge about COVID-19.

**Table 1 ijerph-19-05918-t001:** Sociodemographic characteristics of students.

	Total Number of Students	%
Female	80	71.43
Male	32	28.57
Age group (18–25 years)	103	91.96
Age group (26–35 years)	9	8.04

**Table 2 ijerph-19-05918-t002:** Knowledge about the modes of transmission of COVID-19.

	Correct Answer		95% Confidence Interval	Mean Score ^b^ (Standard Deviation)	95% Confidence Interval [Mean Score]
	Score ^a^	%			
Contact with contaminated surfaces (Yes)	106	94.64	88.80–97.52	4.73 (1.031)	4.54–4.93
Sexual intercourse (vaginal/anal) (No)	62	55.36	46.13–64.23		
Mosquito bites (No)	96	85.71	78.05–91.01		
Person to person (respiratory droplets) (Yes)	110	98.21	93.72–99.51		
Swimming (No)	55	49.11	40.03–58.24		
Shaking hands (Yes)	103	91.96	85.43–95.71		

^a^ Score: the total marks for the questions answered correctly. ^b^ Mean score: the mean calculated from the total marks of the questions.

**Table 3 ijerph-19-05918-t003:** Knowledge about the protective measures against COVID-19.

	Correct Answer		95% Confidence Interval	Mean Score ^b^ (Standard Deviation)	95% Confidence Interval [Mean Score]
	Score ^a^	%			
Wearing face mask (True)	108	96.43	91.18–98.60	7.69 (0.601)	7.58–7.80
Washing hands regularly (True)	110	98.21	93.72–99.51		
Drinking alcohol will kill the virus (False)	110	98.21	93.72–99.51		
Social distancing (True)	109	97.32	92.42–99.08		
Avoid touching eyes, mouth, and nose (True)	111	99.11	95.12–99.84		
Only symptomatic individuals can transmit the virus (False)	104	92.86	86.54–96.34		
Cannot get reinfected (False)	102	91.07	84.34–95.08		
Avoid contact with sick individuals (True)	107	95.54	89.97–98.08		

^a^ Score: the total marks for the questions answered correctly. ^b^ Mean score: the mean calculated from the total marks of the questions.

**Table 4 ijerph-19-05918-t004:** Knowledge about the clinical symptoms of COVID-19.

	Correct Answer		95% Confidence Interval	Mean score ^b^ (Standard Deviation)	95% Confidence Interval [Mean Score]
	Score ^a^	%			
Fever (Yes)	109	97.32	92.42–99.08	7.21 (1.434)	6.94–7.47
Dry cough (Yes)	99	88.39	81.15–93.09		
Tiredness (Yes)	104	92.86	86.54–96.34		
Shortness of breath (Yes)	110	98.21	93.72–99.51		
Vomiting (Yes)	35	31.25	23.41–40.34		
Diarrhea (Yes)	49	43.75	34.92–52.99		
Loss of taste (Yes)	105	93.75	87.66–96.94		
Loss of smell	101	90.18	83.27–94.43		
Headache	95	84.82	77.03–90.30		

^a^ Score: the total marks for the questions answered correctly. ^b^ Mean score: the mean calculated from the total marks of the questions.

**Table 5 ijerph-19-05918-t005:** Students’ perceptions about COVID-19.

	Mean	Standard Deviation (SD)
The lockdown was an effective method of controlling the spread of the disease.	3.92	1.083
It is impractical to practice social distancing during everyday activities. ^c^	3.20	1.400
The virus was created in a laboratory. ^c^	2.85	1.117
The virus originated from bats and spread to humans.	3.11	1.085
I am willing to take the vaccination should any becomes available.	2.60	1.372
I believe the vaccine could have negative side effects and therefore, refuse to take it. ^c^	2.63	1.302

^c^ Reverse scoring for negatively worded questions.

**Table 6 ijerph-19-05918-t006:** Students’ attitudes towards the protective measures implemented.

	Mean	Standard Deviation (SD)
The need to sanitize our hands when we enter the campus or a building is unnecessary. ^c^	4.54	0.967
Only sick/vulnerable people should wear masks on campus. ^c^	4.76	0.713
The university provided us with sufficient COVID-related information.	3.66	1.205

^c^ Reverse scoring for negatively worded statements.

**Table 7 ijerph-19-05918-t007:** Students’ practices of protective measures implemented.

	Mean	Standard Deviation (SD)
I always wear a mask when I am on campus.	4.54	0.657
I do not wear my mask when I am outside the classroom/building. ^c^	3.42	1.242
I practice social distancing on campus and in the classroom/auditorium.	2.02	0.838
I do not practice social distancing on campus when I am with my friends. ^c^	2.73	1.115
I wash my hands more frequently/use sanitizer whenever possible.	4.37	0.684
I attended class even when I had COVID-related symptoms. ^c^	4.23	1.013

^c^ Reverse scoring for negatively worded statements.

## Data Availability

Not applicable.

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
