# Peer review of "Effects of Population Knowledge, Perceptions, Attitudes, and Practices on COVID-19 Infection Prevention and Control in NUST"

_ijerph, 2022, doi:10.3390/ijerph19105918_

Round 1

Reviewer 1 Report

A well-written and relevant paper for the topic. But I think regression should be performed to test the link between variables/factors. Otherwise, the claim "how important it is to have accurate knowledge in relation to the mode of transmission, symptoms, and protective measures of infectious disease" cannot be judged.  The average knowledge rate cannot be an explanatory factor. 

2. The measure for knowledge questions is not accurate enough. Several questions can involve attitude/be equivalent to an evaluation of attitude. While this cannot be changed after the survey, I suggest in future studies, more accurate/in-depth questions which do not involve much attitude element should be adopted.

3. In terms of context, some more description can be done. Why Namibia and what might make difference in the current studies as compared with previous research might be discussed in the Introduction section.    

Reviewer 2 Report

Title: Title must be reivsed in order to reflect the study population considered (University students)

Introduction section

The introduction section must be revised. In particular, more literature review concerning knowledge attitude and practice on COVID-19(or in previous pandemic diseases) infection prevention and control measures should be described in order to better place the study in literature context. Suggested references:

  • Lastrucci, V.; Lorini, C.; Del Riccio, M.; Gori, E.; Chiesi, F.; Moscadelli, A.; Zanella, B.; Boccalini, S.; Bechini, A.; Puggelli, F.; Berti, R.; Bonanni, P.; Bonaccorsi, G. The Role of Health Literacy in COVID-19 Preventive Behaviors and Infection Risk Perception: Evidence from a Population-Based Sample of Essential Frontline Workers during the Lockdown in the Province of Prato (Tuscany, Italy). International Journal of Environmental Research and Public Health 2021, 18 (24), 13386. https://doi.org/10.3390/ijerph182413386.
  • Hong, K.J.; Park, N.L.; Heo, S.Y.; Jung, S.H.; Lee, Y.B.; Hwang, J.H. Effect of e-Health Literacy on COVID-19 Infection-Preventive Behaviors of Undergraduate Students Majoring in Healthcare. Healthcare 2021, 9, 573.

Furthermore, paragraphs concerning COVID-19 diseases description and the related prevention measures are too long and have to be summarized.

Results:

It would be very interesting to repeat all the analyses treating students enrolled in the faculty of health separated as this is a specific group of students that may have different knowledge, attitudes and practices on COVID-19 preventive measures.

Discussion/conclusion

Please clearly report all the limits of the study in a paragraph in the discussion section. Some limitations are described in the conclusion and should be moved in the discussion. Furthermore, non response bias, selection bias  it seems that females students and students from health faculty responded more- should be clearly described.
